

# Snow cover variability and trend over Hindu Kush Himalayan region using MODIS and SRTM data

Nirasindhu Desinayak[1], Anup K. Prasad[1],*, Hesham El-Askary[2], Menas Kafatos[2], Ghassem R. Asrar[3]

[1]Photogeology and Image Processing Laboratory, Department of Applied Geology, Indian Institute of Technology (Indian School of Mines), Dhanbad –826004, India
[2]Center of Excellence in Earth Systems Modeling and Observations, Schmid College of Science and Technology, Chapman University, 452 N. Glassell, Orange, CA 92866, USA
[3]Universities Space Research Association, 7178 Columbia Gateway Drive, Columbia, MD 21046, USA
*Correspondence to*: Anup K. Prasad (anup@iitism.ac.in)

**Abstract.** Snow cover changes has a direct bearing on the regional and global energy and water cycles, and the change in Earth's climate condition The study of long term altitudinal (spatial and temporal, 2000-2017) in the coverage of snow and glaciers in one of the world's largest mountainous region, the Hindu Kush Himalayan (HKH) region including Tibet have been studied using remote sensing data from the Moderate Resolution Imaging Spectroradiometer (MODIS) on Terra (at 5km grid resolution). Terra provided a unique opportunity to study zonal and hypsographic changes in the intra-annual (growing season and melting season) and inter-annual variations of snow and glacial cover over the HKH region (2000-2017). The zonal and altitudinal (hypsographic) analyses were carried out for melting-season and accumulating-season. The altitude-wise linear trend analysis (Pearson's) of snow cover, shown as a hypsographic curve, clearly indicate a major decline in snow cover (average of 5% or more, at 100m interval aggregates) between 4000-4500m and 5500-6000m altitudes, which is consistent with the median trend (Theil-Sen, TS) and the monotonic trend (Mann-Kendall statistics, MK) analysis. The regions and altitudes where major and statistically significant increase (10 to 30%) or decrease (-10 to -30%) in snow cover are identified. The extrapolation of the altitude-wise linear trend shows that it may take between ~74 to 7900 year (for 3001-6000m and 6000-7000m altitude zones respectively) for mean snow cover to decline approximately 25% in the HKH region, assuming no-change in other parameters) that affect the snow cover.

## 1 Introduction

Understanding the impact of snow cover variability with respect to altitude and temperature in the Hindu Kush Himalayan region (HKH) (W. W. Immerzeel et al., 2010; A. Shrestha et al., 2015) is of great importance for regional water availability and understanding of climate change in this highly populated region of the world. The meltwater from snow and ice contribute to all three rivers (Indus, Ganga, and Brahmaputra), with the highest meltwater fraction in the Indus and the lowest in the Ganga (Bookhagen & Burbank, 2010; W. W. Immerzeel et al., 2010; Siderius et al., 2013). The HKH region is projected to show a substantial loss of glacial mass and area in the coming decades (T. Bolch et al., 2012). The snow cover



and glaciers of the high-altitude regions of HKH, including Tibet, are likely to be one of the most affected by the projected rise in global temperatures by about 1–2°C on an average in this decade, which could be significantly higher (reaching 4-5°C) in mountainous terrains (T. Bolch et al., 2012; Prasad et al., 2009; A. B. Shrestha et al., 1999a). The assessment of glacier shrinkage over High Mountain regions of Himalaya and Tibetan Plateau (1960 to 2010), based on a 0.5° grid resolution published data on glacier shrinkage, shows large uncertainties in the rate of shrinkage due to various factors (Cogley, 2016) such rate of snowfall, accumulation and melt.

Snow cover variability is considered as a direct indicator of regional as well as global climate change in the terrestrial domain (Frei et al., 2012; A. Shrestha et al., 2015). Snow cover, as well as the change of its contribution to surface albedo, are listed as Essential Climate Variables (ECV) by the Global Climate Observing System (GCOS). Long-term changes in the thinly snow-covered areas, especially in the mountainous region, affect the albedo and the global radiation budget. GCOS emphasizes that the study of snow cover changes is one of the priority areas that can be achieved using long term satellite observations, especially in remote areas not accessible to any other types of measurement and monitoring. Snow cover fluctuations in HKH regions are highly variable temporally because of various types of controlling factors include topographic effects, glacier dynamics, various types of geomorphic parameters (A. Shrestha et al., 2015), and as of late, due to anthropogenic emissions of soot and other air pollutants (Kang et al., 2019). Global MODIS snow products, also called as the climate-modeling grid (CMG) products at relatively coarser temporal and spatial resolution (5km, monthly, (D.K. Hall et al., 2006)) presents a relatively rapid quantitative analysis of seasonal variability, altitude-wise variability, and temporal trends (spatial and altitude-wise) over the Himalayan region.

## 1.1 Regional warming and decrease in snow cover

The Earth's mean temperature has increased by 0.85 ± 0.2 °C from 1880 to 2012 (Pachauri, 2014). The temperature trends in Nepal (central Himalayas) for the period of 1971-1994 show continuous warming at an average rate of 0.06 °C/year varying spatially and seasonally (A. B. Shrestha et al., 1999b). In contrast, the global average surface temperature rise of the last century was 0.6 ± 0.2 °C (IPCC, 2007). Regional warming is affecting the mountain regions' hydrology due to accelerated cryosphere thawing (Z. Tang & Wang, 2013). Many glacier and snow-covered areas are reportedly decreasing around the world, thereby making the glaciers one of the fastest-changing landforms in the world. Snow cover anomaly over the Himalayan region is of the opposite sign to that over northern Eurasia (Bamzai & Shukla, 1999). The comparative assessment of the Himalayan and Eurasian regions shows that the northernmost regions of both continents are mostly snow-covered during winter and exhibit minimal interannual variability. Eurasian snow cover was high only for one year, 1985, and low for three years, 1975, 1989, and 1990 (Bamzai & Shukla, 1999).

By 2080, the mean warming over India is projected to be 3.3-4.8°C higher relative to pre-industrial times (Chaturvedi et al., 2012). Prasad et al., 2009 reported statistically significant mean month-to-month warming (up to 0.048±0.026°K/year) of the mid-troposphere that is the near-surface atmosphere over the high-altitude Himalayas and Tibetan Plateau. Thus, 2-3 times





higher warming trend (positive and statistically significant) was observed from December to May (accumulation period of snow) compared to the mean positive annual warming trend over the Himalayas (0.016±0.005°K/year) and Tibetan Plateau (0.008±0.006°K/year). The Himalayan and Tibetan Plateau Glacier during this period shows a substantial decrease in snow cover and an extensive glacial retreat (Prasad & Singh, 2007).

## 1.2 Seasonal changes in snow cover

The snow-covered area (SCA), derived from MODIS observations, over the Tibetan plateau, shows not only a decline in the coverage over the period 2003-2010 but also a decreasing tendency in the persistent SCA (SCA > 350 days) also marked by the increase in temperature (0.09 °C/year) and precipitation (0.26 mm/year). However, a slightly increasing tendency was also observed in the maximum SCA over the study area (W. Wang et al., 2015). The seasonal snow cover distribution over Bhutan shows maximum monthly mean snow cover in February and minimum snow cover in July (Gurung et al., 2011). The time series of QuikSCAT radar observations reported by Panday et al., 2011 indicate the beginning of melt onset in late March to early April at elevations of ~4000m, and a delay of approximately one month at higher elevations (>5500m). Freezing begins at the highest elevations (>5500m) around late September and later on with decreasing elevation, with lower elevations (3500–4000m) experiencing freezing around mid-October (Panday et al., 2011).

Over the Tibetan Plateau, the minimum snow cover area occurs in July to August, and the SCA increases rapidly from September, reaching the maximum in March. On average, years 2002, 2005, and 2008 received the largest amount of snow, whereas, in 2001, 2003, 2007, and 2010 received the smallest amount of snow (Duo et al., 2014). The mid-northern hemisphere, especially over Tibetan Plateau, shows a decrease in the number of snow-covered days. Persistent snow was reported to be present in the areas of DEM $\geq$ 4500m, indirectly indicating that the area of persistent snow has decreased over Tibetan Plateau below 4500m  (W. Wang et al., 2015).

## 1.3 Temporal trend of snow cover

Recent studies of temporal trend analysis of snow cover across the Himalayas, and the Tibetan plateau, show a considerable variation. For instance, a significant negative snow cover trend was reported in the upper Indus basins during the winter for the other seasons. The glaciers across the Himalaya were found retreating at a rate of $15.5 \pm 11.8$ m year $^{-1}$ and have lost an overall area of $13.6 \pm 7.9$ % from the last four decades  (W. Immerzeel, 2008; Pratibha & Kulkarni, 2018). A conspicuous variation in the trend from west to east over the HKH has been observed in recent studies. The annual melting rate is conservatively estimated at 1% of the total ice reserve (W. Immerzeel, 2008).

## 1.4 Changes in snow cover with altitude

Recent studies indicate that the trend of snow cover also shows large variations with altitude. For instance, using satellite data Rikiishi & Nakasato (2006) found that the mean annual snow cover area in the Himalaya and the Tibetan Plateau has



decreased by ~ 1 % yr$^{-1}$ during 1966–2001. The rate of decrease is the largest (1.6%) at the lowest elevations (0–500m). On the other hand, the length of the snow-cover season is declining at all elevations, with the highest rate of decline in the 4000–6000 m altitude range. On the Tibetan Plateau (4000–6000m), the length of snow cover season has decreased by 23 days, and the end date for snow cover has advanced by 41 days during 35 years (Rikiishi & Nakasato, 2006). A systematic long term study of selected 19 glaciers in the Chandra-Bhaga basin, Himachal Pradesh (1980–2007) revealed that the Snow Line Altitude (SLA) increased from 5009 ± 61 m to 5401 ± 21 m during this period (Pandey et al., 2012). According to CMIP5 models, snowmelt is projected to occur earlier, while the ice melt component is expected to increase, with considerable ice thinning, and snow cover may disappear below 4000m altitude by the end of the 21st century (Soncini et al., 2015).

## 1.5 Reported analysis of MODIS snow cover data

One of the most useful products for snow cover analysis is the MOD10 snow product developed by the MODIS science team under NASA sponsorship and archived/distributed by the National Snow and Ice Data Center (NSIDC) (D.K. Hall et al., 2006; Dorothy K. Hall & Riggs, 2007). Several previous studies have evaluated the performance of these snow products based on field measurements and reporting less than 10% error for the snow presence (Dorothy K. Hall & Riggs, 2007) and 10% positive bias for the albedo product (B. Tang et al., 2013). Recent studies show that the MODIS snow cover product overestimates the snow cover areas for the Mount Everest region with absolute error ranging from 20.1% to 55.7%, whereas the improved algorithm estimates the snow cover for the HKH region with an absolute accuracy of greater than 90% (B.-H. Tang et al., 2012). The cloud cover and topographic shading in the mountainous regions are known to be the major factors affecting the accuracy of snow cover products. Parajka et al. (2010) developed and validated a regional snow-line method (SNOWL) for estimating snow cover from MODIS daily product, especially during cloudy conditions (up to 90%) over Austria. The methodology provided robust estimates of snow cover over a range of cloudy conditions (10%-90%) for snow-line mapping. Based on MODIS data, Wang and Xie, (2009) proposed a snow cover index (SCI) that can quantify snow cover duration and extent. The SCI analysis for Tianshan Mountains, the least snow cover in the six hydrologic years was reported in August 2005, which affected the perennial snow/glacier cover extent of 2380 km$^2$ area with an elevation of more than 4000m.

The key to understanding the connection between the variation of snow cover and global climate change, especially in the mountainous regions, is the analysis of long term MODIS snow cover data (Wang and Xie, 2009). As the effect of regional warming increases with altitude, understanding the snow cover change with altitude is equally important. Therefore, one of the primary objectives of this current study is to create hypsographic curves to understand the seasonal distribution (snow accumulation and melting season) as well as the temporal trend of snow-covered regions with altitude over the HKH region, in addition to the detailed zonal analysis using long-term MODIS derived snow cover product. The study area comprises of the entire HKH region covering eight countries, namely Afghanistan, Bangladesh, Bhutan, China, India, Myanmar, Nepal, and Pakistan (www.icimod.org).



## 2 Data Sources and Methodology

### 2.1 MODIS Snow Cover

The NASA National Snow and Ice Data Center (NSIDC) Distributed Active Archive Center (DAAC) provides snow cover related data since the launch of the Moderate Resolution Imaging Spectroradiometer (MODIS) onboard *Terra*. In this study, snow cover derived from MODIS sensor onboard *Terra* at a spatial resolution of 5 km (in the gridded form) and as monthly (MOD10CM) composite (version 6) were used to analyze the spatial, temporal as well as attitude-wise variation of snow cover. The snow cover detection and snow cover fraction are based on the Normalized Difference Snow Index (NDSI) algorithm (Hall & Riggs, 2007; Salomonson & Appel, 2004), taking advantage of the contrasting reflectance of snow in the visible and one shortwave infrared bands. The snow albedo is estimated using a radiative transfer model (Klein & Barnett, 2003). Several previous studies have evaluated the performance of these snow products compared to field measurements and reported the error of less than 10% error snow cover presence (Hall et al., 2006; Hall & Riggs, 2007).

### 2.2 SRTM Digital Elevation Model (DEM)

The Shuttle Radar Topography Mission (SRTM) 90m gridded Digital Elevation Model (DEM) version 4.1 was obtained from Consortium for Spatial Information CGIAR-CSI GeoPortal (http://srtm.csi.cgiar.org). These data were resampled to 5 km resolution over the study region and used for the hypsographic analysis of snow cover. The corresponding value of snow cover was extracted from MODIS *Terra* at 5 km to analyze spatial, hypsographic variability, and temporal trends.

### 2.3 Methodology

Monthly average snow cover in 5 km resolution (collection 6, product MOD10CM v6) was obtained in HDF-EOS format from NSIDC DAAC. The data was processed in gridded form, known as Climate Modeling Grid (CMG), from March 2000 to February 2017 and used to extract average snow cover and basic quality assurance (QA) in a Geographic Lat/Lon projection. MODIS pixels were assigned a numeric code for the night (211), cloud (250), no decision (253), water mask (254), and other fill values (255). All the Snow Cover Monthly CMG datasets were extracted for the HKH region, as demarcated by ICIMOD (https://www.icimod.org/). The snow cover extent in each pixel (range of value from 0 to 100) was determined, formatted (ASCII), and archived, and later on were used for the geospatial analysis. The altitude data for the study region were extracted from SRTM DEM and used for generating hypsographic curves and for analysis of snow cover change with altitude and trends. The HKH region was further divided into five different longitudinal zones (west to east) named as zone-1 (60-70° E), zone-2 (70-80° E), zone-3 (80-90° E), zone-4 (90-100° E) and zone-5 (100-105° E). For seasonal analysis, the data were categorized into two seasons, i.e., the Melting season from March to August and the Accumulating season from September to February. The seasonal hypsographic analysis for accumulating and melting seasons were also carried out. For the purpose of comparison, the hypsographic chart also shows the month of February in



the melting season and August in the growing season as a dotted line. The snow cover distribution maps (monthly, yearly and seasonal) and the trend maps illustrating the increase or decrease in snow cover have been prepared. The spatial maps include a geographic location with their hypsographic curve to emphasize snow cover changes with increasing altitude.

## 3. Results and Discussion

### 3.1 Spatial snow cover changes

The mean snow cover (in percent) derived from monthly composites of MODIS *Terra* and its spatial distribution over the HKH region during 17 years (March 2000 to February 2017) is shown in Figure 1a. The average long-term snow cover is found to be 35.9% (Table 1); however, the north-western region (particularly Zone-II) show much higher snow cover as some of the largest glaciers (such as the Siachen Glacier) and permanent snow-covered are within this zone.

The peak in snow cover is observed during February (48.8%), and the lows are observed during August-September (27.9%, and 27.3%, respectively) (Table 1). The statistics for mean snow cover change with altitude, at an interval of 500m, for individual months and the entire study period (March 2000 to February 2017) are shown in Table 1. The highest mean snow cover is found to be above an altitude of 6000 m (96.6%). The long-term average zonal snow cover is found to be 14.4%, 38.7%, 13.5%, 17.2%, and 7.9% for Zone-I, II, III, IV, and V (west to east), respectively (Table 2). The zonal and the altitudinal breakup of mean snow cover show the highest value (99.1%) over the Zone-II (70-80 °E). The Zone-V (100-105 °E) also shows a higher mean snow cover (>96%) even at a relatively lower altitude (5500-6000m) as compared to other zones (Table 3). The white-colored patches in Figure 1a represent some of the major lakes and reservoirs such as Qinghai, Cedo Caka, Selin Co, Nam Co, and Yamzhog Yumco.

The frequency of snow cover (i.e., snow cover persistence) is essential for understanding the temporal variability of snow cover and its spatial distribution. Using the monthly snow cover images, we obtained the frequency of the presence of snow cover (for each pixel) for the entire study duration (204 months), from March 2000 to February 2017. Figure 1b depicts the total number of snow-covered months for the HKH region. The monthly snow cover clearly depicts the high-altitude Himalayan mountain range with permanent snow cover, and also other regions of Tibetan Plateau with relatively permanent snow cover (Figure 1c). Figure 1c shows the regions with persistent snow cover above 80% for the entire study duration, i.e., the permanent snow cover regions in the HKH region based on MODIS observations.

The monthly spatial distribution of MODIS derived mean snow cover (Figure 2a) shows that the maximum and minimum is observed during February and August-September, respectively (Table 1). The monthly average snow cover maps also show that there is an overall gradual increase in the mean snow cover from September to February, which is known as the accumulating season (Figure 2b). Similarly, there is an overall decline from March-August, which is known as the melting season (Figure 2b). The monthly variability over the early phase (2003-04, 2004-2005) and later phase (2011-2012, 2015-16)



compared to the average snow cover over the entire study period (2000-2017) suggest a gradual decline in snow cover from 2004-05 to 2015-16 period. However, specific anomalous years (2003-04 or 2011-12) compared to the entire study period needs to be further investigated in detail. In addition to monthly variability, zonal and altitude variability should be considered in explaining some of these trends and observed anomalies.

## 3.2 Temporal changes in snow cover with altitude

The hypsographic curves of annual mean snow cover for 2000-2016 and the seasonal (March to February) snow cover for 2000-2017 for the HKH region are shown in Figure 3. The mean snow cover is shown as a solid black line (base), and annual lines clearly depict anomalous years compared to the long-term average snow cover. The change in snow cover with altitude clearly shows that there is a sharp jump in snow cover percent between 5001 to 6400m elevations. Above 6400m

altitude, the snow cover is usually very high (>96%) and does not show a large inter-annual variability compared to the range of altitude from 3001-5000m, with overall ~30-35% snow cover. Below 3000m, the snow cover percent gradually declines from ~30% to ~10% up to 1000m altitude. Below 1000m altitude, there is a slight increase in the snow cover (about ~15%) with an increase in interannual variability (compared to 1001-3000m). This increase in interannual variability at lower altitudes may be attributed to known issues such as topographic shadows in the snow cover retrieval algorithm. The

relatively lower seasonal snow cover for 2016 is due to missing data for the months of January and February, which represent the peak of snow accumulating season.

Figure 4a shows the hypsographic curves of seasonal average snow cover during the accumulating (red line), melting (blue line), and all (black line) seasons for the 2000 to 2017 period over the HKH region. It clearly shows that above 5500m altitude, the difference in snow cover between accumulating and melting season reduced with altitude, depicting permanent

snow and glacier-covered regions (>90% snow cover). However, between ~1001 to 5000m altitude, the accumulating, melting seasons and overall data (composites of 100m intervals depicted by red, blue, and black solid line, respectively) show conspicuous differences in the average snow cover (~30% between 3001-5000m and ~10% between 1000-2000m). The shaded region (numerous fine lines of red and blue) indicates individual observations and their very large range and fluctuations between 1001-5000m that gradually decreases at higher altitudes (6000-7000m). The accumulating season

usually has a higher mean snow cover percentage as compared to melting season at the elevations between 800m to 5400m, but below 800m the hypsographic curve behaves oppositely, underlining the issues such as topographic shadows in the snow cover retrieval algorithm.

The hypsographic curve depicting monthly composites of average snow cover during the snow melting and accumulating season over the HKH region (2000-2017) is shown in Figure 4b. In general, the curve is shifting towards the lower mean

snow cover from March to August (snow melting season), whereas shifts towards the higher mean snow cover percentage from September to February (snow accumulating season). For comparison, the maxima are observed in February (in the melting season chart), and the minima are observed in August (in the accumulating season chart), which is highlighted as a



dashed line to mark the end of the previous season. The gradual variation in the monthly mean snow cover with the passage of season, especially between 1001-5000m altitude, is clearly visible. The mean snow cover for the accumulating and melting season is depicted by a solid black line. Above 3000m altitude, the July-August period consistently shows the lowest snow cover during a melting season. Similarly, in the accumulating season, the month of February clearly marks the highest snow cover between 501-6000m altitude. Zonal construction of such hypsographic curves may allow detection of anomalous years that vary significantly from the normal seasonal cycle. Overall, snow cover for the months of December, January, February, March, and April show higher mean values as compared to other months. Above the altitude of 5000, the variability of mean snow cover is found to be minimum. Hysographic analysis (Figure 4b and 4c) shows a notable change from March onwards where substantial decline in snow cover is found (0-5500 m altitude and average black-line) and a increase in snow cover is observed from September onwards (0-5500 m altitude). Melting season is considered from March onwards and growing season from September onwards for such analysis. This is expected as the areal extent of snow cover below 5500m is higher and regions with thin and seasonal snow cover are generally found at relatively lower altitudes which are affected much earlier during the change of season than the thick snow cover/glaciers at higher altitudes. The higher altitude snow cover (5500-7000m) shows a lag (approx. one month) in starting of melting and acculation phase.

### 3.3 Trend Analysis

In general, the overall time series analysis presented here (Figure 5a) shows a very weak and statistically insignificant negative trend (or decline) of snow cover over the entire study area. This in contrast to the several studies and databases, based on ground and satellite observations, over last 2-3 decades ((Berthier et al., 2007; Tobias Bolch, 2007; Kulkarni, 2006; Raup et al., 2007) that shows contrsting changes in snow cover over the Himalayas. This implies that regional and altitudinal analyses are required to capture the variability (Figure 5a). The line chart with a linear fit shows the time series analysis of monthly mean MODIS snow cover over the HKH region (2000-2016) (Figure 5a). The variation of pixel count (area) is shown as histograms. The snow cover is usually found to be lowest in the months of July or August and highest in January or February, respectively.

The pixel-level trend analysis (linear fit) at 5km grid resolution, based on the monthly time series of mean snow cover for the entire HKH region (Figure 5b), show large spatial variability, as calculated from the slope of the linear fit. Figure 5c present the statistically significant (95% confidence interval) pixels or aggregates of pixels (regions) where the linear trend is strongly positive or negative (-30 to 30%). The central Himalayan mountains, northern parts of Nepal show a significant increase in snow cover (red and brown color zones, 10 to 30%), whereas parts of Arunachal Pradesh and eastern Himalayan and Tibetan Plateau show a significant and large decline in the snow cover (green color zones, -10 to -30%). Snow and glacier images from Arunachal Pradesh show a large decline in snow cover and signs of glacier melting (Prasad & Singh, 2007). For example, a recent study (Basnett et al., 2013) of changes in the area of glaciers in Tista Basin occupying ~200Km$^2$, Sikkim, Eastern Himalayas show a loss of 3.3 ± 0.08% in the area and an increase in the debris-covered area by



6.5±1.4 km$^2$ during 1989-90 and 2010 timeframe. Light yellow color regions (Figure 5c) represent statistically significant regions where the linear trend is found to be negative (up to -10%).

The hypsographic curve depicting total changes in snow cover for the entire HKH region for 2000-2017 period, obtained from Linear (LIN) fit, is shown in Figure 6a. The thick solid line (red/black) represent aggregates of values at 100m interval, and thin red/black horizontal bars represent percentage changes at 1m interval. The red color is altitudinal variation of total

255 changes in snow cover for statistically significant regions only, whereas the black line is for the entire region (including both statistically significant and insignificant areas). The red line, for statistically significant regions, shows larger variability and clearly indicates an overall loss in snow cover with altitude. The total change in snow cover (100m aggregates) is found to be about -5% for altitudes of 4001-4500m and 5501-6000m. However, the thin horizontal red lines indicate that the statistically significant loss for specific regions can vary and be even higher, i.e., -5 to -15% (Figure 6a).

Figure 6c presents the hypsographic curve depicting total changes in snow cover over for entire HKH region during 2000-2017, obtained from the Linear trend (LIN, Pearson product-moment linear correlation), and the median trend (TS, Theil-Sen). The variability of rate of change of snow cover, based on these analyses, are largely similar over the range of altitude for relatively long time series (17 years). The mean rate of change (LIN method) shows a peak between 4001-4500m (about -2.5%) and 5501-6000m (about -3%) altitudes. We obtained a similar linear trend from median trend analysis (TS) and the

monotonic trend (Mann-Kendall (MK) statistics. The Z score is shown in the same scale as LIN and TS for comparison) in Figure 6b. Overall, the trend analysis (LIN, TS and MK, Figure 6b) clearly shows a large reduction in glacier cover area only for 4001-4500m and very high-altitude regions (5501-6000m). The statistically significant regions, as in linear trend analysis (Figure 6a), clearly show a large decline in snow cover (average of 5% or more, at 100m intervals) between 4001-4500m and 5501-6000m altitudes. The mean snow cover percent change is found to be -1.0% and -2.0% in Zone 4 and 5, respectively

(Table 4). The altitudinal change in snow cover shows that only a limited part of the HKH region located in the 6001-7000m elevation (between Zone II-IV, between $70°$-$100°E$, Table 5) is found to be positive, thereby implying that no decline in snow cover is observed in very high altitude regions (above 6000m).

Table 5 is the summary of the overall zonal trend analysis for the entire study period (2000-2017). The change in snow cover is presented as an average at 500m intervals. The regions at altitudes of 3501-4500m and 5001-6000m show a decline in

snow cover, but this decline varies widely for different attitudes and zones (west to east). For example, the westernmost zone (Zone 1, 60°-70°E) extends up to 5000m altitude but does not show any decline in snow cover. The adjacent Zone 2 (70°-80°E) shows a decline in snow cover (-0.13% between 5501-6000m altitude whereas no such decline is observed at very high altitude 6001-7000m. The central region (Zone 3, 80°-90°E), that extends up to 7000m altitude, shows more signs of decline in snow cover between 3001-4500m and 5001-6000m elevations, whereas no such decline is observed at higher

altitudes 6001-7000m. Further east, Zone 4 (90°-100°E), shows greater signs of the overall decline in snow cover from 2501-6000m elevation, whereas no such decline is observed at higher altitudes 6001-7000m. The eastern zone (Zone 5) is most





affected, showing a decline in snow cover at all altitudes (501-5500m) except at the highest altitude in this region (5501-6000). The largest decline is found to be around -11.2% between 4501-5000m altitude in zone V.

The observed linear trend, for altitudes between 3001-7000m at an interval of 500m, suggests that it may take hundreds of years for any drastic change in the mean snow and glacier cover, especially at higher altitudes 6001-7000m (Figure 7). The altitudinal linear trend for 3001-7000m, at 500m interval for the study period (2000-2017) shows that it may take 76-94 year for approximately 25% decline in the mean snow cover for altitudes of 3001-4500m and 5501-6000m, and hundreds or thousands of year for 4501-5500m and 6001-7000m altitudes, over the HKH region (Figure 7).

## 4. Summary and Conclusions

We found spatial and altitudinal changes in snow and glacier cover during 2000-2017 for the Hindu Kush Himalayan (HKH) region. There are significant heterogeneity and variability in these changes due to the vast area that this region covers. We conducted a zonal and altitudinal analysis of these changes using a variety of statistical analyses. We found;

1. Zonal (western, central, and eastern) variations in snow cover are prominent where the highest snow and glacier cover is found in the western zones. Zone I and Zone II, located between 60°-80°E, show snow cover at 46.4% and 48.2%,
respectively.

2. The variation in snow cover with altitude, calculated at 500m intervals, is not uniform across the HKH region. Zone II, III, IV, and V, located between 70-100°E, show greater than 93% snow cover between 6001-7000m altitude, and between 5501-6000m in Zone 5.

3. The mean snow cover for the study period (2000-2017) during melting or accumulating seasons generally remains
below 40% up to ~5500m altitude but rapidly increases to >90% at higher altitude (6000-7000m).

4. In general, the mean snow cover during the study period (2000-2017) was found to be below 30% for 3001-5000m altitude during the melting season (March to August), but it is greater than 30% during the accumulating season (September to February).

5. The average decreasing rate of snow cover/ glacier area is very low (-0.0024 % per year, based on linear trend analysis
(LIN), and is insignificant for the entire region. However, the HKH region show large and contrasting regional variations both across the zones (west to east), and also with altitude.

6. The central and eastern zones (Zone III, IV and V, 80°-105°E) of the HKH show more prominent decline in snow cover as compared to Zone I and II (60°-80°E). In contrast, the western Zone-1 and Zone-II contain specific regions where the snow cover trend is found to be positive (increase during the study period).

7. At higher altitudes, particularly between 6001-7000m, all zones (Zone-II, III and IV) show an increasing trend, compared to a contrasting declining trend at relatively lower altitudes (5501-6000m).



8. The declining trend of snow cover is observed at increasingly lower altitudes and moving eastwards from Zone III to Zone V. Zone 5 shows a negative trend at all altitudes, except at the highest altitude (5501-6000m).

9. The linear trend analysis for altitudinal changes suggest that it may take 76-94 year for 25% decline in the mean snow cover for 3001-4500m and 5501-6000m altitudes, and perhaps hundreds of years for 4501-5500m, and 6001-7000m altitudes, over the HKH region.

10. Though the increasing trend is observed over specific regions, a substantial decline in some specific zones/regions observed from temporal trend analysis is a cause for concern.

It is observed that better long-term datasets, with the higher spatial and temporal resolution, will provide greater insight for regions that show rapid change.

**Acknowledgments**

The authors are grateful to NASA National Snow and Ice Data Center (NSIDC) Distributed Active Archive Center (DAAC) for providing snow cover related data since the launch of Moderate Resolution Imaging Spectroradiometer (MODIS) onboard *Terra*. The authors are thankful to Consortium for Spatial Information CGIAR-CSI GeoPortal for providing the Shuttle Radar Topography Mission (SRTM) 90m gridded Digital Elevation Model (DEM) version 4.1 used in this study.

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





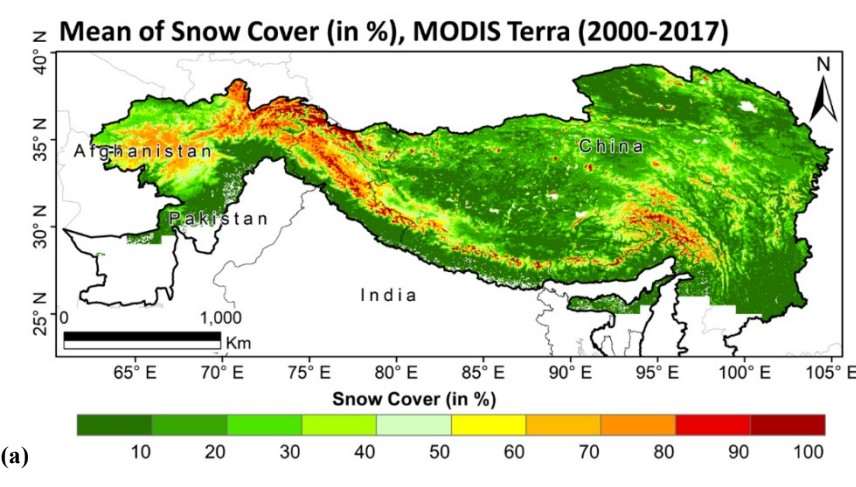

**(a)**

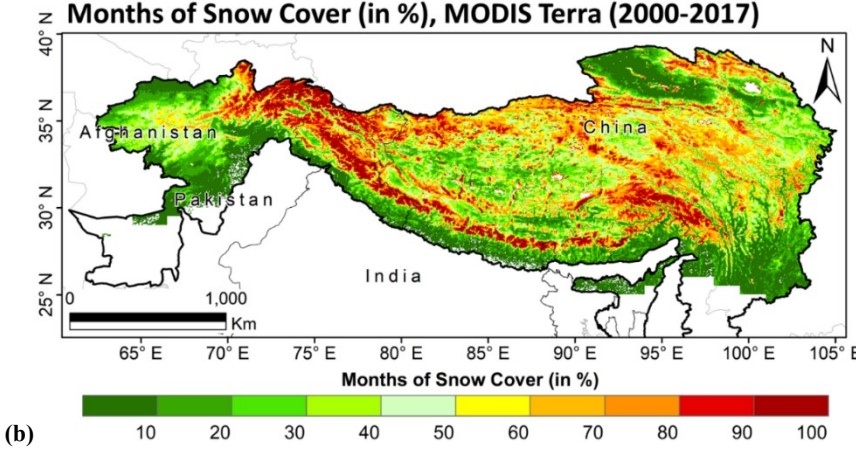

**(b)**

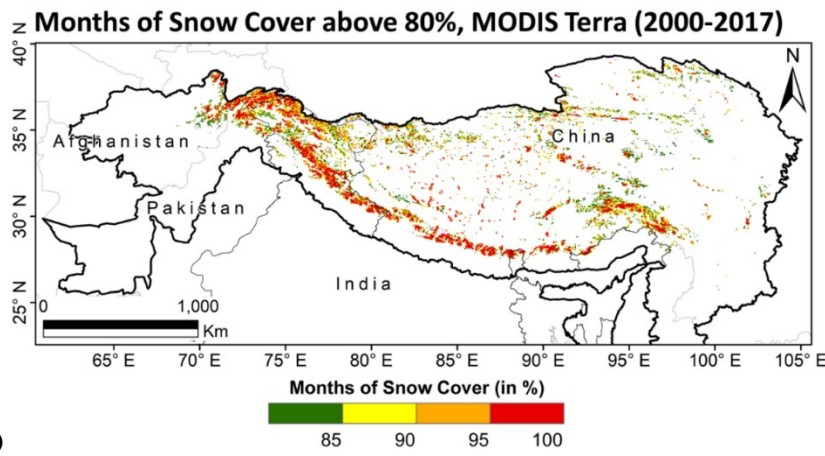

**(c)**

**Fig. 1. The spatial distribution of MODIS Terra derived (a) monthly mean snow cover (in percent), (b) the total duration of the presence of snow cover (in percent) out of 204 months, and (c) the presence of snow cover above 80% (total months out of 204 months) during March 2000 to February 2017 over the HKH region**







**Fig. 2. The monthly spatial distribution of MODIS derived snow cover (in percent) over the HKH region during 2000-2017: (a) monthly maps, (b) monthly variability over the years and (c) monthly variability over the early phase (2003-04, 2004-2005) and later phase (2011-2012, 2015-16), compared to average for the entire study period (2000-2017).**





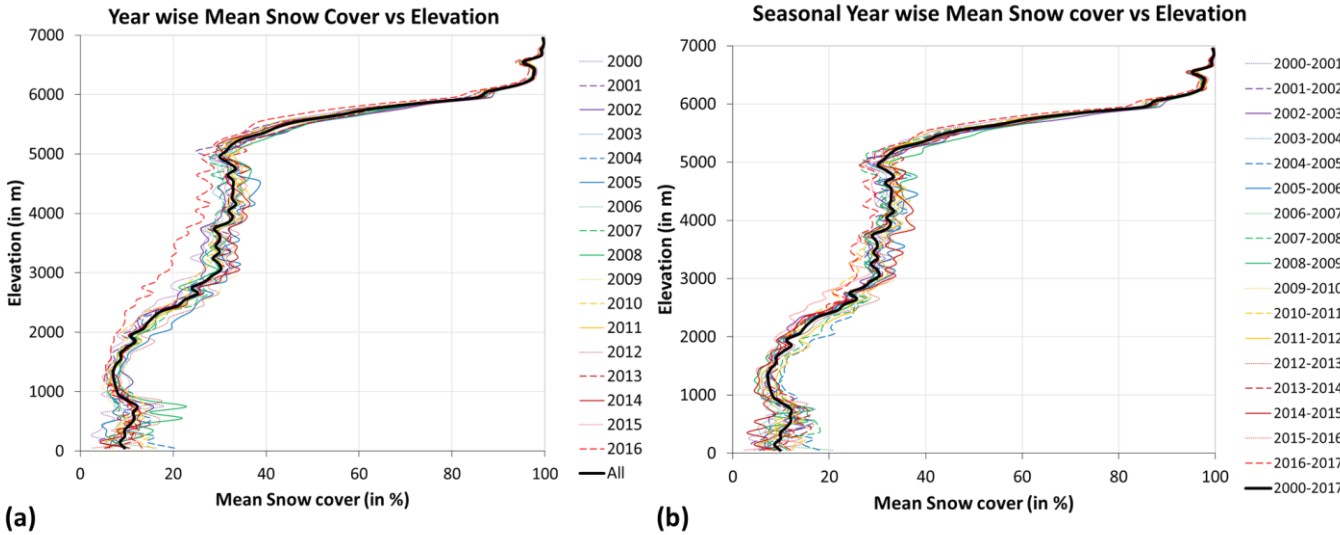

**(a)** **(b)**

**Fig. 3. The hypsographic curves of (a) year-wise mean snow cover (in percent) for 2000-2016 and (b) Seasonal year-wise (March to Feb) snow cover (in percent) for 2000-2017, respectively, over HKH region. Note, the seasonal average of snow cover for year 2016 is lower because of missing data for the months of January and February.**







**Fig. 4. (a) The mean snow cover hypsographic curves during 2000 to 2017 over HKH region: (a) seasonal curves for accumulating season (red line), melting season (blue line) and all season (black line), (b) monthly composites (in percent) during the snow melting and accumulating seasons. (c) The variability of mean snow cover over months (at different altitudes, <5500m) show starting of decline in snow-cover from the month of March onwards (black line is average).**





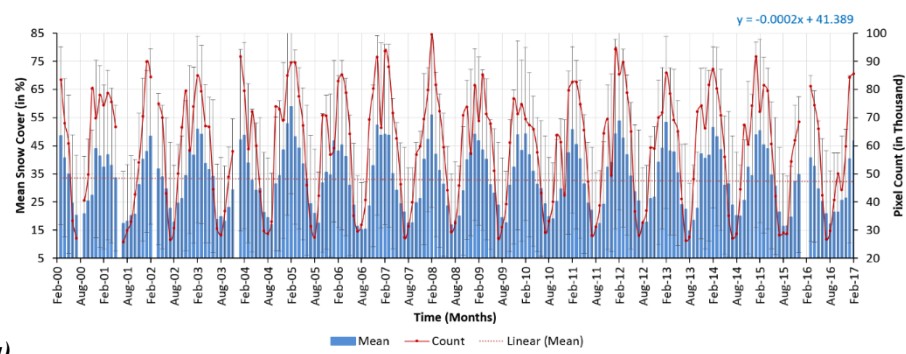

*(a)*

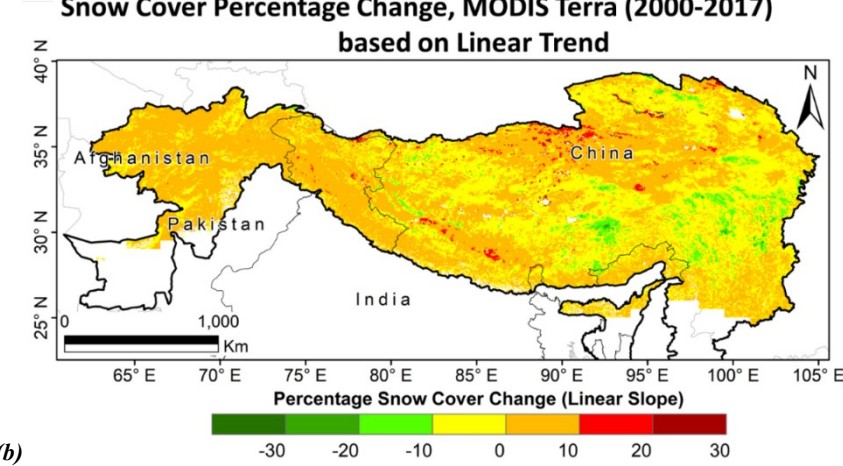

*(b)*

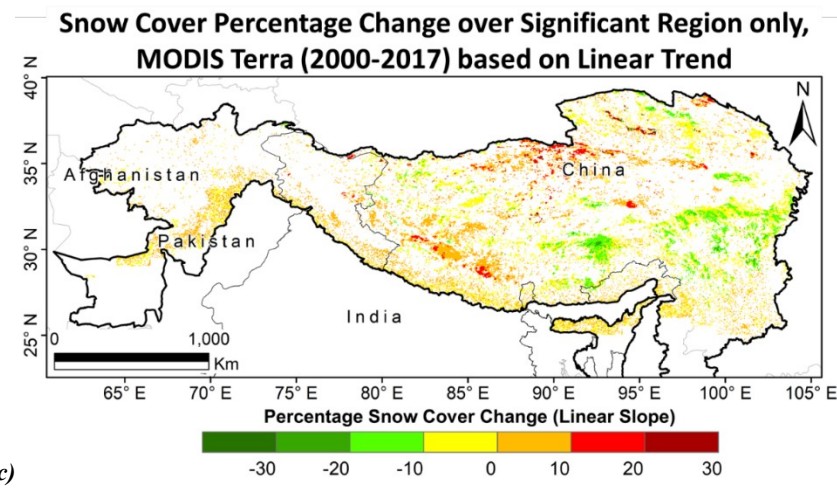

*(c)*

**Fig. 5.(a) The time series analysis of monthly mean MODIS snow cover during 2000 to 2016, over the HKH region. The snow cover**
**variability (percentage change) was derived from monthly MODIS Terra data during 17 years (March 2000 to February 2017) for**
**(b) the entire HKH region, and (c) statistically significant sub-regions of HKH region only.**





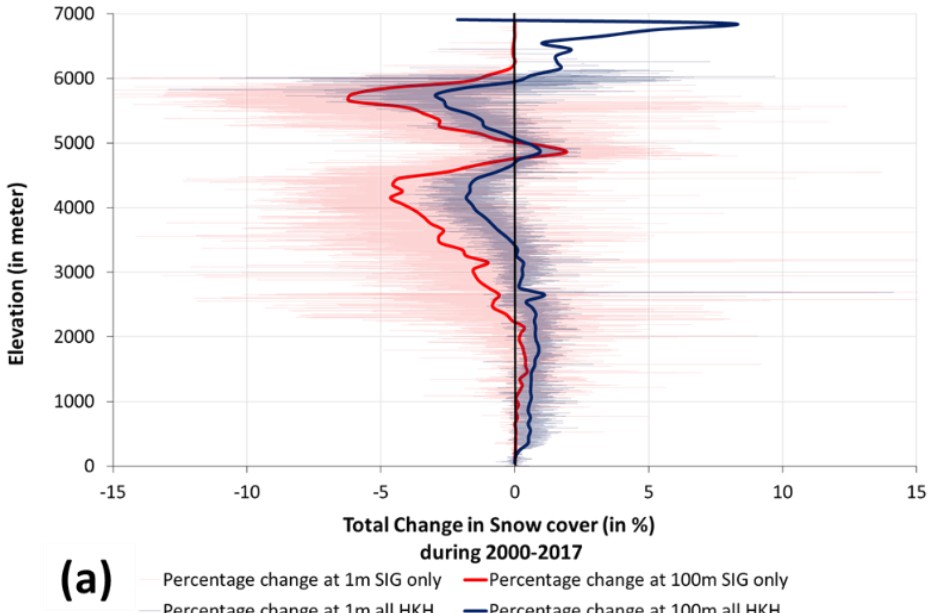

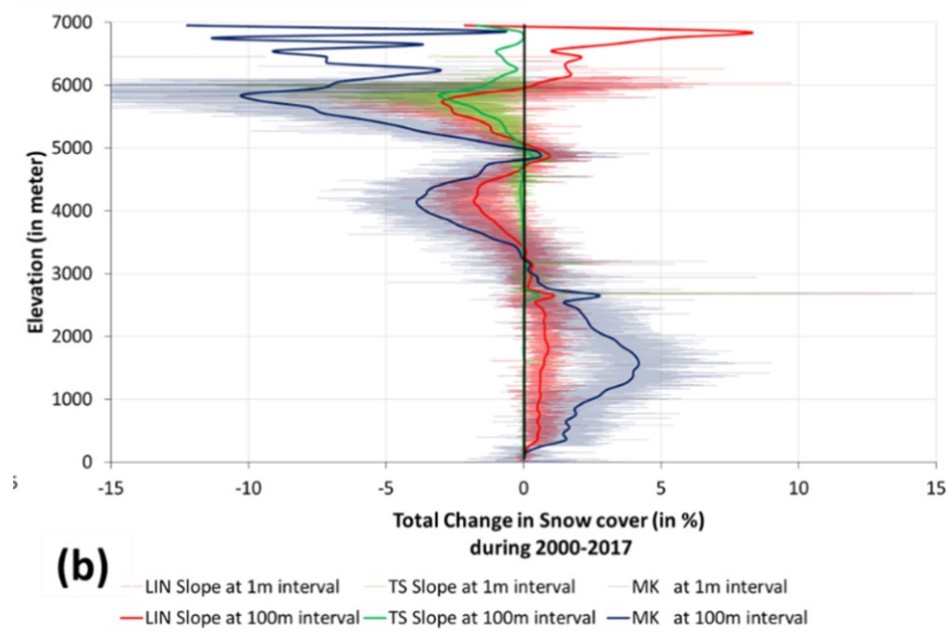

**Fig. 6. The altitude-wise trend analysis (hypsographic curves) for: (a) Snow cover change (in percent) derived from the linear fit (pearson product moment correlation, LIN) over the entire HKH region (black curve) and statistically significant trend regions only (red curve), (b) The linear fit model (red curve) over the HKH region (LIN) is compared with the median trend (Theil-Sen, TS) and the monotonic trend (Mann-Kendall statistics, MK, Z-score) analysis.**




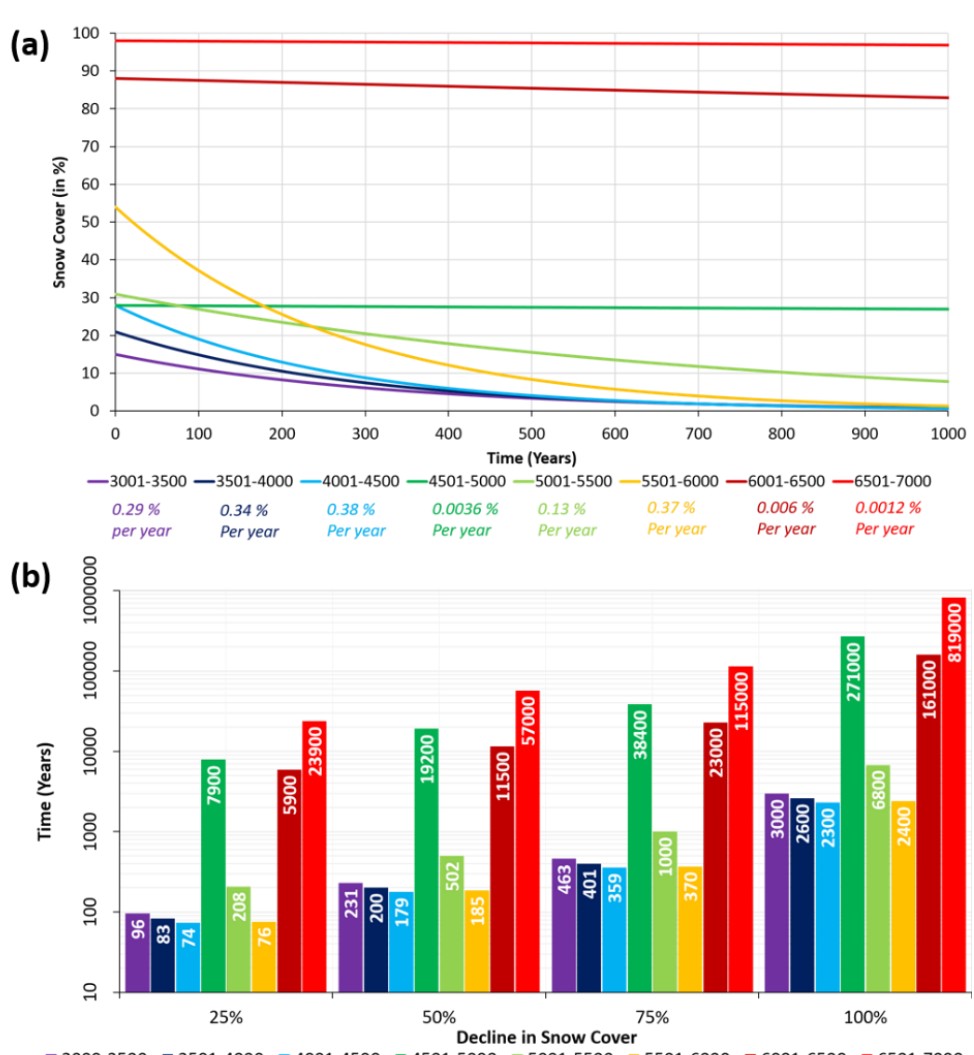

**Fig. 7. The results from linear trend analysis of Snow Cover change (mainly reduction) during 2000-2017, and in subsequent years, over the HKH region and where the trend was found to be statistically significant (95% CI).**





**Table 1. Mean snow cover (%) for different altitudes, derived from composites of individual months for the study period ( March 2000 to February 2017) over the HKH region.**

| Elevation | 0-500 | 501-1000 | 1001-1500 | 1501-2000 | 2001-2500 | 2501-3000 | 3001-3500 | 3501-4000 | 4001-4500 | 4501-5000 | 5001-5500 | 5501-6000 | 6001-6500 | 6501-7000 | Average |
|---|---|---|---|---|---|---|---|---|---|---|---|---|---|---|---|
| **Aug** | 15.46 | 13.11 | 5.68 | 7.97 | 11.01 | 12.18 | 14.21 | 15.07 | 16.45 | 18.3 | 20.11 | 51.4 | 91.69 | 98.03 | 27.91 |
| **Sep** | 10.88 | 10.94 | 5.55 | 6.08 | 8.08 | 10 | 10.16 | 11.15 | 15.62 | 18.31 | 25.78 | 56.86 | 94.03 | 98.44 | 27.28 |
| **Oct** | 3.71 | 2.81 | 1.97 | 2.85 | 3.79 | 6.7 | 9.29 | 16.5 | 24.55 | 31.03 | 38.43 | 63.86 | 95.94 | 98.83 | 28.59 |
| **Nov** | 2.98 | 3.91 | 4.11 | 5.7 | 9.56 | 15.69 | 24.65 | 31.23 | 37.97 | 40.08 | 41.31 | 62.1 | 95.3 | 98.77 | 33.81 |
| **Dec** | 6.32 | 12.97 | 13.3 | 15.12 | 25.92 | 41.8 | 47.17 | 43.22 | 41.46 | 39.96 | 38.85 | 58.79 | 93.22 | 97.98 | 41.15 |
| **Jan** | 8.47 | 16.16 | 16.33 | 23.36 | 39.91 | 55.7 | 54.23 | 46.67 | 43.53 | 38.74 | 39.53 | 63.67 | 93.16 | 98.75 | 45.59 |
| **Feb** | 8.46 | 19.72 | 18.2 | 26.86 | 47.27 | 65.98 | 55.46 | 47.04 | 43.19 | 39.49 | 43.61 | 73.68 | 95.3 | 98.77 | 48.79 |
| **Mar** | 2.33 | 3.66 | 5.86 | 10.39 | 20.34 | 42.59 | 52.15 | 44.95 | 43.11 | 41.53 | 45.41 | 78.71 | 97.3 | 99.41 | 41.98 |
| **Apr** | 2.83 | 2.49 | 3.41 | 5.58 | 9.53 | 21.38 | 35.07 | 39.29 | 40.87 | 38.37 | 44.41 | 77.92 | 97.62 | 99.5 | 37.02 |
| **May** | 5.26 | 4.59 | 3.07 | 4.91 | 5.86 | 13.3 | 22.23 | 30.76 | 34.5 | 31.58 | 40.22 | 71.66 | 96.62 | 99.06 | 33.12 |
| **Jun** | 18.51 | 19.99 | 6.53 | 7.22 | 9.7 | 13.32 | 18.68 | 25.71 | 30.79 | 27.02 | 31.38 | 62.17 | 94.08 | 97.97 | 33.08 |
| **Jul** | 32.21 | 22.18 | 7.69 | 8.55 | 11.72 | 14.44 | 17.54 | 19.98 | 23.44 | 23.3 | 25.52 | 52.37 | 91.23 | 97.49 | 31.98 |
| **All Months** | 9.79 | 11.04 | 7.64 | 10.38 | 16.89 | 26.09 | 30.07 | 30.96 | 32.96 | 32.31 | 36.21 | 64.43 | 94.62 | 98.58 | 35.86 |

**Table 2. Zonal analysis of yearly mean snow cover (%) for 2000 to 2016 period.**

| Years | Zone - 1 | Zone - 2 | Zone - 3 | Zone - 4 | Zone - 5 |
|---|---|---|---|---|---|
| **2000** | 11 | 36 | 13 | 19 | 11 |
| **2001** | 12 | 38 | 12 | 16 | 9 |
| **2002** | 14 | 38 | 16 | 17 | 7 |
| **2003** | 15 | 40 | 13 | 16 | 6 |
| **2004** | 13 | 40 | 11 | 18 | 8 |
| **2005** | 17 | 42 | 16 | 21 | 11 |
| **2006** | 19 | 42 | 15 | 17 | 8 |
| **2007** | 17 | 35 | 13 | 16 | 8 |
| **2008** | 15 | 38 | 14 | 21 | 10 |
| **2009** | 17 | 43 | 13 | 18 | 8 |
| **2010** | 11 | 39 | 11 | 15 | 6 |
| **2011** | 14 | 38 | 13 | 17 | 8 |
| **2012** | 18 | 39 | 14 | 18 | 9 |
| **2013** | 15 | 39 | 16 | 18 | 7 |
| **2014** | 15 | 39 | 14 | 17 | 7 |
| **2015** | 15 | 42 | 16 | 15 | 6 |
| **2016** | 6 | 30 | 9 | 14 | 6 |
| **All Yrs** | 14.4 | 38.7 | 13.5 | 17.2 | 7.9 |





**Table 3. Zonal and Altitudinal (at 500m interval) distribution of Snow Cover (%) over the HKH region.**

| Elevation (in m) | SCP_Z1 (60°-70°E) | SCP_Z2 (70°-80°E) | SCP_Z3 (80°-90°E) | SCP_Z4 (90°-100°E) | SCP_Z5 (100°-105°E) | All Zone HKH |
|---|---|---|---|---|---|---|
| 0-500 | 17.30 | 2.53 | 7.56 | 8.25 | | 8.91 |
| 501-1000 | 24.93 | 2.73 | 3.41 | 6.20 | 14.84 | 10.42 |
| 1001-1500 | 16.57 | 4.71 | 2.21 | 4.84 | 8.69 | 7.40 |
| 1501-2000 | 15.83 | 13.97 | 5.40 | 7.33 | 5.72 | 9.65 |
| 2001-2500 | 32.31 | 29.49 | 5.50 | 10.00 | 7.28 | 16.92 |
| 2501-3000 | 56.29 | 47.11 | 17.00 | 10.70 | 8.61 | 27.94 |
| 3001-3500 | 67.22 | 56.11 | 21.50 | 14.04 | 10.25 | 33.82 |
| 3501-4000 | 72.92 | 61.63 | 17.33 | 19.46 | 16.39 | 37.55 |
| 4001-4500 | 77.83 | 63.08 | 7.46 | 21.53 | 26.28 | 39.24 |
| 4501-5000 | 82.50 | 60.47 | 26.78 | 29.35 | 48.57 | 49.53 |
| 5001-5500 | | 58.32 | 28.67 | 46.80 | 85.59 | 54.85 |
| 5501-6000 | | 76.45 | 81.67 | 75.55 | 98.49 | 83.04 |
| 6001-6500 | | 98.44 | 97.00 | 93.41 | | 96.28 |
| 6501-7000 | | 99.94 | | 96.26 | | 98.10 |
| | 46.37 | 48.21 | 24.73 | 31.69 | 30.06 | 37.01 |





**Table 4. The zonal changes in snow cover (in percent), derived from the linear fit (LIN).**

| Zone Name | Pixel Count (Snow Cover Area) | Minimum SCPC | Maximum SCPC | Mean SCPC | STD |
|-----------|------------------------------|--------------|--------------|-----------|------|
| Zone-1 | 15297 | -8.62 | 11.44 | 1.45 | 2.20 |
| Zone-2 | 19516 | -30.04 | 71.71 | 1.44 | 3.51 |
| Zone-3 | 33368 | -23.99 | 76.41 | 0.34 | 4.45 |
| Zone-4 | 49343 | -98.30 | 97.98 | -0.99 | 5.76 |
| Zone-5 | 17624 | -26.19 | 42.54 | -1.97 | 4.11 |

**Table 5. The zonal changes in snow cover  (in percent) at 500m elevation interval, derived from the linear fit (LIN).**

| Elevation (in m) | Zone 1 (60°-70°E) | Zone 2 (70°-80°E) | Zone 3 (80°-90°E) | Zone 4 (90°-100°E) | Zone 5 (100°-105°E) | All Zone HKH |
|------------------|-------------------|-------------------|-------------------|--------------------|--------------------|--------------|
| 0-500 | | 0.04 | 0.00 | 0.02 | | 0.32 |
| 501-1000 | 2.12 | 0.12 | 0.05 | 0.02 | -1.45 | 0.64 |
| 1001-1500 | 1.80 | 0.87 | 0.22 | 0.29 | -1.28 | 0.66 |
| 1501-2000 | 1.08 | 2.04 | 0.44 | 0.40 | -0.32 | 0.84 |
| 2001-2500 | 0.75 | 1.90 | 0.68 | 0.18 | -0.11 | 0.77 |
| 2501-3000 | 1.08 | 0.88 | 0.54 | -0.31 | -0.33 | 0.34 |
| 3001-3500 | 2.86 | 0.11 | -0.02 | -1.27 | -0.45 | 0.22 |
| 3501-4000 | 4.64 | 0.56 | -1.00 | -1.70 | -1.66 | -0.76 |
| 4001-4500 | 4.85 | 0.46 | -0.23 | -1.50 | -6.38 | -1.37 |
| 4501-5000 | 3.65 | 1.50 | 1.16 | -0.01 | **-11.18** | 0.39 |
| 5001-5500 | | 1.12 | -0.46 | -3.63 | -4.27 | -0.94 |
| 5501-6000 | | -0.13 | -2.65 | -2.26 | 3.75 | -1.89 |
| 6001-6500 | | 3.84 | 2.05 | 2.02 | | 2.43 |
| 6501-7000 | | **9.91** | 2.83 | 2.54 | | 4.62 |

