# Peer review of "Snow cover variability and trend over Hindu Kush Himalayan region using MODIS and SRTM data"

_Annales Geophysicae, 2021_

## Author Response (AR1)

**RC1**

**Authors:** Authors are thankful to the reviewer for the thoughtful comments. The suggestions to include some of the relevant and recent work in this area has been incorporated in the revised manuscript. The pointwise response related to data/plots, limitations of snow cover algorithm, and cloud cover issues, shadows are given below. We are hopeful that the Reviewer and Editor would find the updated content responsive to the valuable comments/suggestions by the reviewer.

**Review status**: this preprint is currently under review for the journal ANGEO.

**Snow cover variability and trend over Hindu Kush Himalayan region using MODIS and SRTM data**

**Nirasindhu Desinayak et al.**

**Status**: open (until 31 Jul 2021)

**Comment types**: **AC** – author | **RC** – referee | **CC** – community | **EC** – editor | **CEC** – chief editor | : Report abuse

- **RC1**: 'Comment on angeo-2021-29', Anonymous Referee #1, 28 Jun 2021

  This work requires more rigor. The authors are requested to modify the manuscript as per the following comments.

  Page 1:

  1. i) Do the authors think the period 2000-2017 can be considered long enough to be termed "long term"? Since the focus is on snow cover the authors are requested to be clear wherever they mention glaciers.

     Response: The use of space-based Earth observation data records in this study can be considered as "relatively long-term". The manuscript, including the abstract section, has been modified accordingly. In terms of climate change, we agree that 30-year or even 100-year datasets may be more illuminating, but the fact that even this decadal record suggests accelerated change is valuable and makes a case for continuity of these observations. Because the remote sensing coverage for global snow cover monitoring is relatively shorter than this, the present study is an effort to provide analysis over this highly sensitive region to make a case for greater attention to the impact of climate on snow cover globally. We envision that this analysis can be repeated in the future to assess the validity of our findings and conclusions with considerably longer-term datasets.

     ii) Why did the authors choose to use coarser resolution MODIS in presence of higher resolution Landsat data?

Response: We used this unique and only available dataset over such a large and remote region (Himalayan and Tibet region) of the world to understand the regional and altitude-wise changes in the snow cover. This is the best and available dataset in the grid format, as mentioned in the manuscript and known as the Climate Modeling Grid (CMG) dataset, that captures changes over the study area. With this regional perspective, one may now choose to focus on the areas showing significant changes (hotspots/anomalies) to delineate and better understand these changes in greater detail using higher resolution snow cover data, including mass balance studies. We have now suggested such a need/prospect for future research to the revised manuscript.

iii) The authors are requested to consider rewriting the Abstract for better readability.

Response: We agree with the reviewer, and we have made some changes to improve the readability of the abstract.

iv) The authors present some important figures with large variation (like 74-7900 years). This is quite a large deviation to be considered good for a scientific prediction. Please define clearly what are the "other parameters" which are assumed to be unchanged. How is it justified to consider a "no-change" situation?

Response: The extrapolation and projections of relatively long-term trend from the satellite data show that the changes occurring over the Himalayas are varying to a large extent (spatially and altitude-wise). The linear trend is dependent on historical data, the contributing factors, processes, and feedback mechanisms, assuming they stay the same moving into the future. In reality, the current and past contributing factors may or may not be the same in future, thus our projection/extrapolation should be viewed with caution; however, they do provide some insight and what to watch for going forward. In the relevant sections of the manuscript, the details of linear trend and its extrapolation (in such no-change situations originating from linear trend based on historical data and conditions only) are clarified. The authors agree with the reviewer that projecting future changes in snow cover is considerably more difficult and complex than using a linear-trend analysis but it is a starting point for illustrating a greater need for observations and understanding of this highly sensitive region of the world with so many people depending on this source of freshwater resources. As suggested by the reviewer, we have modified the sentence for better readability.

Page 2:

2.  i) The authors are requested to cite the following article where they mention the "anthropogenic emissions of soot…"

-Gautam et al., "Satellite observations of desert dustâ induced Himalayan snow darkening", Geophysical Research Letters, 2013.

Response: We are thankful to the reviewer for this suggestion. The suggested article has been included in the manuscript.

ii) In the section discussing the "Regional warming and decrease in snow cover", the authors are requested to separate the discussion between the changing state of snow cover and glaciers over the HImalayan region. Furthermore, it is difficult to summarize the information presented in this section and the idea still appears quite vague.

Response: We agree with the reviewer that the region's warming and related studies are important. We have attempted to summarise this issue with some recent papers in this sub-section (1.1 Regional warming and decrease in snow cover). We've added two more references (Duan and Wu 2006; You et al. 2017) that provide greater detail about the warming and cloud cover issues and their impact. The updated summary, as well as the pertinent material contained in the cited references, will provide an overview of the research conducted on the region's warming. Because the current study focuses on the fluctuation of snow cover, the authors expect that the summary supplied with more references will suffice given the manuscript's emphasis on snow cover.

Page 3:

3. i) The authors are requested to separate the increasing temperature and precipitation since it is counterintuitive to visualize that both together cause decrease in persistent SCA.

Response: Based on published research in this area, some observed variations in the snow cover and associated changes over the Tibetan plateau between 2003 and 2010 have been cited. As suggested by the reviewer, we have modified the sentence for better readability in the revised manuscript.

ii) The authors are requested to cite the following article, which discusses the seasonal variation in snow cover and its altitudinal trend, in the section discussing the "Seasonal changes in snow cover".

-Muhuri et al., "Snow cover mapping using polarization fraction variation with temporal RADARSAT-2 C-band full-polarimetric SAR data over the Indian Himalayas", IEEE JSTARS, 2018.

Response: We are thankful to the reviewer for this recent article based on SAR data. The suggested article has been included in the suggested section of the manuscript.

Page 4:

4. i) Line 105-111: The authors are requested to cite a recent work discussing the performance of the snow cover mapping algorithms in mountainous areas affected by forests and topographic shading.

-Muhuri et al., "Performance Assessment of Optical Satellite Based Operational Snow Cover Monitoring Algorithms in Forested Landscapes", IEEE JSTARS, 2021.

Response: We are thankful to the reviewer for suggesting this latest article. The suggested article has been included in the manuscript.

Page 5:

5. i) The authors are requested to introduce terrain shadow masks in their analyses. Terrain shadow changes as a function of the time of the year due to solar elevation angle variation. These are the regions of ambiguity. Shadow masks will provide a more robust touch to this work.

Response: The data, its quality, robustness, and broad limits are summarised in the sub-section (1.5 Reported analysis of MODIS snow cover data). As mentioned in the manuscript, the cloud cover and topographic shading in the mountainous regions are known to be major factors affecting the accuracy of snow cover products. The inclusion of such dynamic shadow masks may further improve the snow cover algorithms and the corresponding datasets, especially in higher resolution datasets (<500 m). This may be considered as beyond the scope of the present analysis (based on MODIS data). In this study, the fill values (such as for cloud) have been taken care of during data processing.

Page 8:

6. i) Fig 5 a): The authors are requested to plot the elevation range wise snow cover extent histogram as plotted in the following article,

Response: We agree with the reviewer that information regarding "elevation range wise snow cover extent" would be useful to readers. Please note that the desired data regarding "elevation range wise snow cover extent" is already provided in Table 1 and 2. Table 1 also provides a month-by-month breakdown of this data, and Table 2 shows the variability of such data by zone.

-Muhuri et al., "Snow cover mapping using polarization fraction variation with temporal RADARSAT-2 C-band full-polarimetric SAR data over the Indian Himalayas", IEEE JSTARS, 2018.

7. ii) In the trend analysis how did the authors take into account the errors in the snow detection algorithm? How did the authors deal with partially or completely cloud covered conditions? How did the authors handle the snow cover in forested areas? There is little discussion regarding these issues in the manuscript.

Response: We agree with the reviewer that these are all important contributing factors that affect remotely sensed observations of such a large and remote area. The process of converting top of atmosphere radiance measurements to Earth surface-based geophysical parameters, such as snow cover, takes into account as much as possible the impact of these contributing factors. Quite often, field campaigns are conducted to better characterize and account for such contributing factors, and the remaining effects are included/reported as errors associated with such products.

The MODIS 5km snow cover dataset is a widely used/referenced product that captures very well elevation-wise spatial variability and temporal trend in the snow cover as a function of time (i.e., several months to years). The data, its quality, robustness, and broad limits are summarised in the sub-section (1.5 Reported analysis of MODIS snow cover data). As mentioned in the manuscript, the cloud cover, forest cover, and topographic shading in the mountainous regions are known to be major factors affecting the accuracy of snow cover products. Despite these limitations, the MODIS snow cover data has been demonstrated to be reliable in various studies. The cited references (including the suggested references by the reviewer) discusses the potential contributions of factors identified by the reviewer.

**Citation**: https://doi.org/10.5194/angeo-2021-29-RC1

**Authors:** Authors are thankful to the reviewer for the comments. We have revised the manuscript as per the suggestions. We are hopeful that the Reviewer and Editor would find the updated content satisfactory.

**Authors:** Authors are thankful to the reviewer for the thoughtful comments. The suggestions to include changes such as a common unit for temperature (°C) has been incorporated in the revised manuscript. The pointwise response related to plots, limitations of snow cover algorithm, and cloud cover issues, shadows are given below. We hope that the Reviewer and Editor will find the revised manuscript responsive to the reviewer's remarks and suggestions.

**Status**: final response (author comments only)'

**Snow cover variability and trend over Hindu Kush Himalayan region using MODIS and SRTM data**

**Nirasindhu Desinayak et al.**

**RC2**: 'Comment on angeo-2021-29', Anonymous Referee #2, 24 Aug 2021

I agree with my colleague making a comment on 28th June 2021 that this paper require more additional work. Especially, I see a problem with having only 17 years long series that are further used for deriving trends (or even extrapolating what can happen in 7 000 years (Fig. 7 and rows 285-290)).

Response: In this study, the use of space-based Earth observation data records (17 years) can be considered as "relatively long-term" in terms of identifying major findings. We agree that 30-year or even 100-year datasets are more informative in terms of climate change, but the fact that even this decadal record shows rapid change is noteworthy and provides a case for the observations' continuity. The current work is an attempt to offer analysis across this extremely sensitive region in order to establish a case for increased attention to the influence of climate on snow cover globally. As the satellite-derived dataset expands, future studies can utilise even longer-term datasets to evaluate the validity of our findings and conclusions. We have made this point more clear in the revised paper that continued focus on observation, research and modelling analysis in this highly sensitive region of Earth is warranted.

Generally, some papers cited in the work are rather old (before 2000) - would be better to have newer references (if possible). The same is true for description of observed trends (e.g. row 51).

Response: The cited articles cover the major findings with respect to region's snow cover and warming trend. We have attempted to further amplify this point with some recent papers in this sub-section (1.1 Regional warming and decrease in snow cover). We've added two more references (Duan and Wu 2006; You et al. 2017) that provide greater detail about the warming and cloud cover issues and their impact. The

updated summary, as well as the pertinent material contained in the cited references, will provide an overview of the research conducted on the region's warming, and support the findings of this study. We hope that the summary provided in the revised manuscript together with supporting references will suffice to address the issue of warming the reviewers raised, given this manuscript's focus is on dynamics of snow cover.

"Why °K and °C are used - I think just °C would work better for the whole paper.

Regarding trends - authors use trend year, but per decade may be better (and sound more robust).

Response: The units have been changed to °C, as suggested by the Reviewer, to provide consistency in reading. In the relevant sections (introduction and results and discussion), the trend related to snow cover has been reported/cited as per year basis. The results of the study have been reported as per year basis for consistency in reading. However, based on the data used, the trend per decade can be also discerned and reported at least as a benchmark for future studies. The readers should be able to see how annual and almost two decades of changes relate in terms of magnitude and trend.

Generally, I am missing at least a small discussion about results - there is only description of the results (in parts 3 and 4), with quite complicated description in part 3, but no discussion on it. It should asnwer at least the question, how the length of the analyzed data can influent the results? And how did the authors handle with possible errors / problems in snow detection algorithm (deep valleys, clouds, forests ...)?

Response: For ease of reading, section 3 has been broken into sub-sections with pertinent headings. The known issues related to the MODIS snow-detection algorithms, as also mentioned by the Reviewer, has been discussed in the data and methodology section as well as results and discussion section with relevant citations.

The description of some figures is not sufficient (what are abscissae in Figs. 2c or 5a).

Response: In figure 2c, the abscissa (x-axis) is given as time in months. The ordinate and abscissa in 5a is also labelled as Northings (25°-40° N) and Eastings (65°-105° E).

If data from Jan and Feb 2016 is missed – wouldn't be better to omit it from the Fig. 3a - and how was it handled in other analyses?

Response: As mentioned by the Reviewer, the missing data is clearly shown in the figure 3a. As the data analysis spans a much longer period (17 years), the missing data

of (few months in this case) is not likely to greatly influence the outcome from the analysis (spatial, altitudinal, and temporal trend analysis).

Fig. 4a – there are some suspicious values around 2700-2800 metres – where do they come from and are they correct?

Response: As the fill-values have already been removed from the dataset, the average values of mean snow cover in percentage (at 100m interval) are shown along with average values (at 1m interval) as calculated from the MODIS data. These values are correct and depict large variability in snow cover with increase in altitude. We believe this is an important aspect and finding of this study that have been made clearly now.

**Citation**: https://doi.org/10.5194/angeo-2021-29-RC2

**Authors:** Authors are thankful to the reviewer for the invaluable comments. We have revised the manuscript as per the suggestions. We are hopeful that the Reviewer and Editor would find the updated content satisfactory.

---

## Author Response (AR2)

**Report #1**

**Authors:** Authors are thankful to the reviewer for the thoughtful comments. As suggested, we have included a brief description of trend analysis with suitable references. We are hopeful that the Reviewer and Editor would find the updated content responsive to the valuable comments/suggestions by the reviewer.

I think the revision made many points much more clear.
What I am still missing a bit, is more insight into trend analysis based on the 18years long period of observation - this should be mentioned and discussed more detailed in the text.

Response: As mentioned by the reviewer, we have now added a short description about the linear trend analysis (LIN, TS and MK) in 2.3 Methodology section along with relevant references.

As for chapter 3 - the changed structure is better for readiness, but my previous point regarding discussion of the results was not reflected - as can be see in pdf with tracked changes ...
I am not sure about the point "Does the author give proper credit to related work and does he/she
indicate clearly his/her own contribution?" - I cannot see any reactions from authors' side.
Response: The conclusion section summarizes the findings over the Himalayan region which is in consonance with earlier studies as mentioned in the manuscript and cited references. Authors had tried to highlight altitude-wise variability and trend analysis of snow cover using suitable graphs and tabular data to emphasize importance of the region in terms of significant changes in snow cover in recent decades that warrant further attention and analysis of snow cover dynamics vis-à-vis climate change (temperature change).

**Authors:** Authors are thankful to the reviewer for the invaluable comments. We have revised the manuscript as per the suggestions. To make colour figures friendly and accessible for readers with colour vision deficiencies (CVD), we have now updated the figures with necessary adjustments in colour using the given colour simulator and as per the guidelines of the journal,

We are hopeful that the Reviewer and Editor would find the updated content satisfactory.

**Report #2**

**Authors:** Authors are thankful to the reviewer for the suggested corrections and thoughtful comments. The formatting of data and units have been made to be consistent along with other additions/corrections as suggested by the reviewer. We are hopeful that the Reviewer and Editor would find the updated content responsive to the valuable comments/suggestions by the reviewer.

Dear authors,

here are some comments that I think will improve the quality of the manuscript:

Line 43: The year of the citation should be in brakets ("2021")
Response: We have made the correction in the revised manuscript, as suggested by the reviewer.

Line 44: "Except for one exception" should be "With one exception", if that is what you meant.
Response: We have made the correction in the revised manuscript, as suggested by the reviewer.

Line 55: Two times "various types", I suggest to remove it in this line. And don't you mean geomorphologic parameters instead of geomorphic?
Response: We have made the change in the revised manuscript, as suggested by the reviewer.

Line 59: Rewrite the sentence to "... at relatively coarser temporal (monthly) and spatial (5km) resolution (Hall, Riggs, ...".
Response: We have revised the sentence in the revised manuscript, as suggested by the reviewer.

Line 63 (onwards): I would suggest using the unit Kelvin (K) for temperature according to the International System of Units (SI). Especially when you describe trends or developments, "K/year" is much more common the "°C/year". Also, the format should be consistent over the whole manuscript (sometimes there is a blank, sometimes none).
Response: We are thankful to the reviewer for this suggestion and we have made the formatting consistent (number and its unit, without space) across the manuscript. As most of the cited studies had used °C for the temperature related data, we have chosen to it keep the same for consistency throughout the manuscript.

Line 85: Add a "+" sign to both values.
Response: We have made the correction in the revised manuscript, as suggested by the reviewer.

Line 98: Use "elevation" instead of "DEM".
Response: We have made the correction in the revised manuscript, as suggested by the reviewer.

Lines 103-104: Adapt font size.
Response: We have made the correction in the revised manuscript, as suggested by the reviewer.

Line 147: Don't you mean "altidtue-wise variation"?
Response: We have made the correction in the revised manuscript, as suggested by the reviewer.

Line 159: Redundant information, use either "collection 6, product MOD10CM" or "MOD10CM v6".
Response: We have made the correction in the revised manuscript, as suggested by the reviewer.

Line 162: You should also explain the other possible values ranges of the product, also explain which layer from the Hdfdataset you are using.
Response: We have added the name of layer and range of values in the revised manuscript, as suggested by the reviewer.

Line 171: Better use "accumulation season" instead of "growing".
Response: We have made the correction in the revised manuscript, as suggested by the reviewer.

Line 211 onwards: I suggest to use the term "snow coverage" when directly speaking of percentages.
Response: We have shown snow cover in percentage for consistency across the manuscript.

Line 220: I suggest using "... decreases with altitude" instead of "reduced with altitude".
Response: We have made the correction in the revised manuscript, as suggested by the reviewer.

Line 240-241: Better use "it is found that the variability is minimal".
Response: We have kept the suggested sentence as it is.

Line 251: Remove one of the doubled brackets.
Response: We have made the correction in the revised manuscript, as suggested by the reviewer.

Line 263: Use "km²" instead of "Km²" and add space.
Response: We have made the correction in the revised manuscript, as suggested by the reviewer.

Line 280: Use the same format as above "(MK, Mann-Kendall statistics; ...)".
Response: We have made the correction in the revised manuscript, as suggested by the reviewer.

Line 334: I would not use the word "better" since longer time series are requested.
Response: We have made the correction in the revised manuscript, as suggested by the reviewer.

Line 545: Can you introduce the abbreviation SCPC?
Response: We have made the correction and added the abbreviation for SCP in the revised manuscript, as suggested by the reviewer.

If you incorporate these changes, I'll see the manuscript ready for publication.

**Authors:** Authors are thankful to the reviewer for the list of suggested changes. We have revised the manuscript as per the suggestions. To make colour figures friendly and accessible for readers with colour vision deficiencies (CVD), we have now updated the figures with necessary adjustments in colour using the given colour simulator and as per the guidelines of the journal,

We are hopeful that the Reviewer and Editor would find the updated content satisfactory.